# Collection of cancer-specific data in population-based surveys in low- and middle-income countries: A review of the demographic and health surveys

Chukwudi A. Nnaji[1,2]*, Jennifer Moodley[1,2,3]

**1** School of Public Health and Family Medicine, Faculty of Health Sciences, University of Cape Town, Cape Town, South Africa, **2** Cancer Research Initiative, Faculty of Health Sciences, University of Cape Town, Cape Town, South Africa, **3** SAMRC Gynaecology Cancer Research Centre, University of Cape Town, Cape Town, South Africa

* nnjchu001@myuct.ac.za

## Abstract

Population-based surveys, such as those conducted by the Demographic and Health Surveys (DHS) Programme, can collect and disseminate the data needed to inform cancer control efforts in a standardised and comparable manner. This review examines the DHS questionnaires, with the aim of describing and analysing how cancer-specific questions have been asked from the inception of the surveys to date. A systematic search of the DHS database was conducted to identify cancer-specific questions asked in surveys. Descriptive statistics were used to summarise the cancer-specific questions across survey years and countries. In addition, the framing and scope of questions were appraised. A total of 341 DHS surveys (including standard, interim, continuous and special DHS surveys) have been conducted in 90 countries since 1985, 316 of which have been completed. A total of 39 (43.3%) of the countries have conducted at least one DHS survey with one or more cancer-specific questions. Of the 316 surveys with available final reports and questionnaires, 81 (25.6%) included at least one cancer-specific question; 54 (17.1%) included questions specific to cervical cancer, 41 (13.0%) asked questions about breast cancer, and 8 (2.5%) included questions related to prostate cancer. Questions related to other cancers (including colorectal, laryngeal, liver, lung, oral cavity, ovarian and non-site-specific cancers) were included in 40 (12.6%) of the surveys. Cancer screening-related questions were the most commonly asked. The majority of the surveys included questions on alcohol and tobacco use, which are known cancer risk factors. The frequency of cancer-specific questions has increased, though unsteadily, since inception of the DHS. Overall, the framing and scope of the cancer questions varied considerably across countries and survey years. To aid the collection of more useful population-level data to inform cancer-control priorities, it is imperative to improve the scope and content of cancer-specific questions in future DHS surveys.

**Data Availability Statement:** DHS survey questionnaires are publicly available as part of final reports published and freely accessible on the DHS Programme's website (https://dhsprogram.com/

Methodology/Survey-Types/DHS-Questionnaires.
cfm). All data underlying the findings reported in
this review are presented in the manuscript.

**Funding:** The authors received no specific funding
for this work.

**Competing interests:** The authors have declared
that no competing interests exist.

# Introduction

The burden of cancer incidence and mortality is rapidly growing worldwide, accounting for
an estimated 10 million deaths and about 20% of all deaths globally in 2020 [1]. The burden is
disproportionately high in low- and middle-income countries (LMICs), where access to cancer
care is often sub-optimal [2]. Particularly, the burden of cancer-related mortality is signifi-
cantly higher in LMICs, accounting for 65% of all cancer deaths globally despite a lower inci-
dence of cancer compared with high-income countries (HICs) [2–4]. The share of global
cancer deaths occurring in LMICs is projected to increase to 75% by 2030 [4]. Due to the prev-
alence of carcinogenic infections like *Helicobacter pylori*, hepatitis B virus, and human papillo-
mavirus, LMICs bear a significant burden of infection-associated cancers, including gastric
cancer, hepatocellular carcinoma and cervical cancer, in contrast to HICs where these types of
cancers are less common.[2] National and subnational cancer data are useful tools for assessing
the magnitude of cancer burden and informing cancer control priorities [5, 6]. Yet, there is
very limited availability of high quality cancer data in many LMICs, with only one in five coun-
tries able to report cancer data of sufficient quality to determine incidence and prevalence esti-
mates [5, 6].

Population-based surveys offer a potentially useful source of data to address data gaps, pro-
viding a key source of information about the prevalence, risk factors and patterns of diseases
and public health issues [7, 8]. In the context of cancer control, such information may assist in
guiding appropriate cancer control interventions, track changes over time, and support the
evaluation of cancer programmes. Population-based surveys, such as those conducted by the
Demographic and Health Surveys (DHS) Programme, can collect and disseminate the data
needed to inform cancer control efforts in a standardised and comparable manner [9, 10].
Since 1984, the DHS programme has collected, analysed, and disseminated nationally-repre-
sentative data across various themes of population health in over 90 LMICs [11]. The surveys
cover topics, such as maternal and child health, HIV/AIDS, malaria, nutrition, women's
empowerment, fertility, and family planning. DHS Surveys usually involve a large number of
respondents (between 5,000 and 30,000) and typically are conducted about every 5 years, to
allow comparisons over time [11].

The DHS programme supports countries with standard survey methodologies, manuals,
and procedures for data collection, including standard model survey questionnaires. While the
DHS surveys are designed to be comparable across countries, each country can adapt its survey
to suit its national context and health information needs, enabling the collection of contextu-
ally relevant and useful data for informing country-level data-informed decision making. For
instance, DHS data and findings are commonly used to benchmark country-level progress and
evaluate potential impacts of specific health policies and interventions [9]. Readily available
and representative data are needed to inform cancer control priorities, evaluate the perfor-
mance of cancer interventions and strengthen cancer control programmes [10]. Population-
based surveys may be particularly important in LMICs where reliable cancer registries and
high quality health information systems are often lacking [12].

Although the DHS questionnaires have historically included questions on cancer risk fac-
tors, prevalence, awareness and screening, little is known about the nature and extent to which
cancer-specific questions have been asked over time and across country contexts. An impor-
tant question is whether the DHS questionnaires currently allow for the collection of data that
are useful for informing and strengthening cancer control programmes in LMICs, or whether
there are opportunities for making the questions more relevant and fit-for-purpose.

Therefore, this study examines DHS questionnaires since inception of the surveys in 1984
to 2022, with the aim of reviewing and critically evaluating the types of cancer-specific data

collected by the DHS in LMICs from the inception of the surveys to date. Specifically, it aims to identify the cancer-specific questions and to assess the frequency, nature and scope of those questions. For the purposes of this assessment, cancer-specific questions refer to those relating to specific aspects of cancer, such as survey participants' knowledge and awareness of particular cancers; prevalence of commonly occurring cancers; and awareness, practices and trends relating to cancer screening, diagnosis and treatment. As a secondary objective, this study aimed to quantify and characterise questions specific to alcohol consumption and tobacco use, which are known risk factors of cancer and other non-communicable diseases.

Ultimately, the study seeks to provide an evidence base for guiding the design and implementation of future DHS and similar population-based surveys to more effectively collect context-appropriate data related to cancer awareness, prevention, screening, prevalence, early diagnosis and treatment practices. Such data will be useful for supporting national cancer control programmes and efforts, by providing nationally-representative and context-specific data on priority cancers. Furthermore, revising the DHS to better integrate cancer-related questions can help support and accelerate the World Health Organization's Global Monitoring Framework on non-communicable diseases (NCDs) and other global efforts aimed at tracking progress in the prevention, early detection and control major NCDs including cancers in LMICs [10].

## Methods

### Study design

This study applied a systematic approach to searching, identifying and analysing DHS data collection tools for cancer-specific questions asked since the inception of the surveys to the date of search.

### Structure of DHS surveys

DHS surveys collect primary data using four types of model questionnaires: a household questionnaire used to collect information on characteristics of the household's dwelling unit and characteristics; individual woman's questionnaire, individual man's questionnaire and the biomarker questionnaire is used to collect biomarker data on children, women, and men. Individual men and women questionnaires include information on fertility, mortality, family planning, marriage, reproductive health, child health, nutrition, and specific diseases such as malaria, HIV/AIDS, tuberculosis, breast cancer and cervical cancer. In addition to these, individual questionnaires often include questions on common cancer risk factors like tobacco use and alcohol consumption. Country-specific questions are typically added to meet local conditions and needs. Periodic updates to the model questionnaires are made through an open consultation and invitation of input from the public, to meet countries' existing and emerging data needs. For special information on topics that are not contained in the model questionnaires, optional questionnaire modules are available. Survey questionnaires are made publicly available as part of final reports published on the DHS Programme's website [11].

### Search strategy

A systematic search of the DHS database was conducted to identify cancer-specific questions asked during the surveys from inception of the DHS programme in 1984 to August 2022, and updated in December 2022 [13]. Survey questionnaires were searched for questions relating to specific aspects of cancer, such as survey participants' knowledge and awareness of particular cancers; prevalence of commonly occurring cancers; and awareness, practices and trends

relating to cancer screening, diagnosis and treatment. The search was conducted in three steps. The first step involved searching the DHS Programme's website using the website's 'search by survey characteristics' search function, which specifies keywords such as cancer, breast cancer, cervical cancer, prostate cancer, and screening to identify survey questionnaires that included questions on these topics. A filter was applied to restrict the search to DHS surveys (standard, interim, continuous and special DHS surveys) while excluding other survey types such as the AIDS Indicator Survey (AIS), Malaria Indicator Surveys (MIS) and Service Provision Assessment (SPA). From the initial search results, ten surveys were randomly selected (two for every decade from the 1980s). The second step of the search involved a full text assessment of the questionnaires of these surveys (usually available as appendices in the full reports of each survey) to identify relevant cancer-related search terms for the next step of the search.

In the third step of the search, assessments of the full texts of all available survey questionnaires were conducted using the search terms identified from the previous step. Such search terms included: cancer, tumour, screening, breast, breast cancer, breast examination, mammography, cervical cancer, cervix, Pap smear, Human papillomavirus, lung cancer and prostate cancer. To identify surveys with questions specific to alcohol consumption and tobacco use, which are known risk factors of cancer, the DHS database's 'search by survey characteristics' search function was applied to identify all surveys in which alcohol and tobacco use questions were asked since inception of the DHS. Search terms were adapted for French, Spanish and Portuguese texts. All available survey questionnaires since the inception of the DHS programme were searched, with no language restrictions. Questionnaires in any language other than English were translated using a web-based translation tool [14].

### Data extraction

Cancer-specific questions identified from the third step of the search and other relevant information were extracted using a data extraction tool, and charted under the following domains identified in the second step of the search: cancer risk factors; prevalence; awareness and knowledge; screening and detection; treatment and follow-up; and barriers to cancer services.

### Data analysis

Descriptive statistics were used to quantitatively summarise the number of cancer-specific questions and questionnaires across survey years and countries. Historical trends of surveys and cancer-specific questions were illustrated using time-series charts, while a heatmap was used to show the global distribution of the frequency of surveys asking cancer-specific questions across countries. Narrative summaries were presented by cancer type and emergent themes. In addition, the framing, scope and depth of identified survey questions were appraised.

## Results

The search returned a total of 341 DHS surveys (including standard, interim, continuous and special DHS surveys) conducted in 90 countries since 1985. As of the last search on 19 August 2022, 316 of these surveys have been completed with final reports and reports available, while the rest are ongoing. Of the 316 surveys with available final reports and questionnaires, 81 (25,6%) included at least one cancer-specific question; 54 (17.1%) included cervical cancer-specific questions, 41 (13.0%) asked questions about breast cancer, and 8 (2.5%) included questions related to prostate cancer. Questions related to other cancers (including laryngeal, liver, lung, oral cavity, ovarian and non-site-specific cancers) were included in 40 (12.6%) of the surveys. In terms of geographical spread, a total of 39 (43.3%) of the 90 countries where DHS surveys have been conducted to date have at least one survey with one or more questions focused

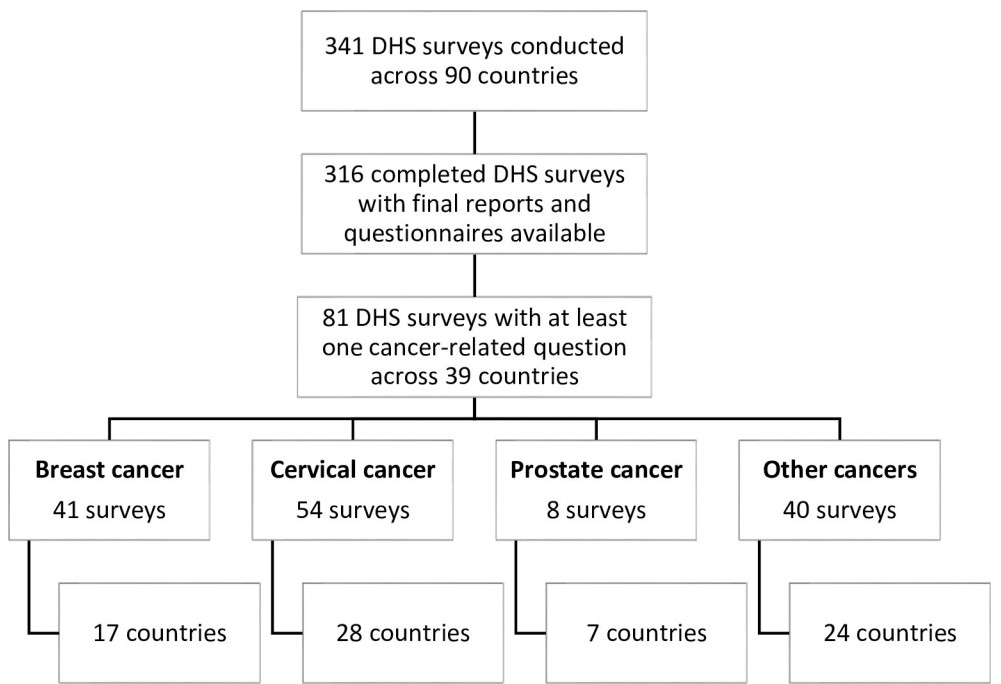

**Fig 1. Distribution of cancer-specific questions across DHS surveys, by cancer types and country settings.**

on cancer. Cervical cancer-specific questions were asked in more countries (27), compared with breast (17 countries) and prostate cancer (8 countries), Questions related to other or non-site specific cancers were asked in 24 countries. Fig 1 illustrates the search results and the distribution of cancer-specific questions by cancer sites and country settings.

Fig 2 shows the trends of the number and proportions of DHS surveys with cancer-specific questions over time. Overall, the number and proportions of surveys with cancer-specific questions increased, though unsteadily, across the years from 1985, with the proportions increasing from around 15% in the earliest period (1985–1989) to 60% in 2020–2021. Table 1 shows the distribution of cancer-specific questions by cancer type, country and survey year.

Fig 3 is a heatmap illustrating the global distribution of DHS surveys with cancer-specific questions by frequency across countries. The spatial trend shows countries in the Latin America and the Caribbean (LAC) region having conducted more surveys with cancer-specific questions than those of other regions. Of the countries with two or more surveys collecting questions on cancer, seven are in the LAC region (Bolivia, Brazil, Colombia, Dominican Republic, Guatemala, Honduras and Peru), while three are in Africa (Lesotho, Namibia and South Africa, all of which are in southern Africa); another three are in Asia (India, Jordan and the Philippines) and two are in Europe (Albania and Armenia). While the majority of LAC countries that have conducted at least one DHS survey have asked cancer-specific questions in one or more of those surveys, the opposite is the case in Africa, with the majority of countries in the region having not asked cancer related questions in their DHS surveys despite the high number of DHS surveys conducted on the continent.

## Cervical cancer

Questions on cervical cancer (often referred to as cancer of the neck of the womb in surveys) were asked in 54 surveys across 28 countries. Cervical cancer screening questions featured the

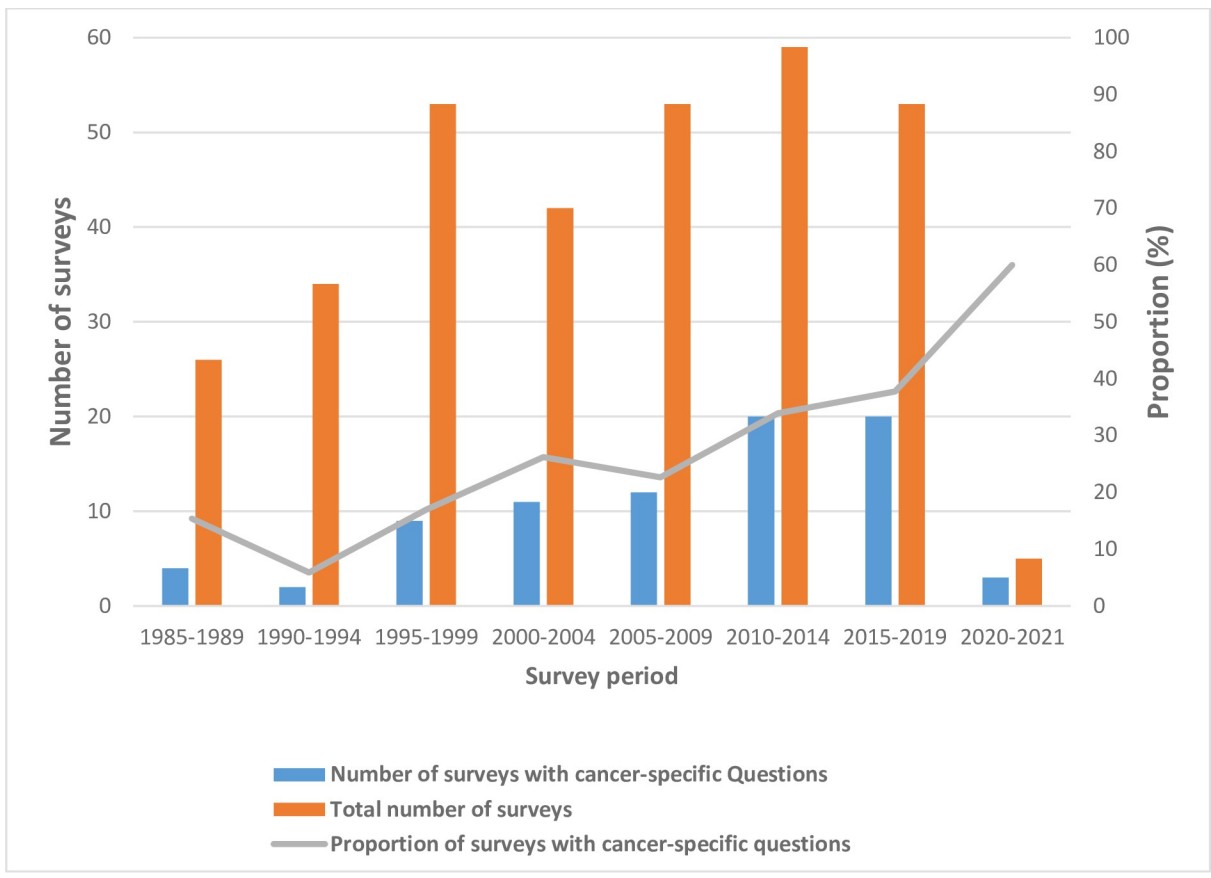

**Fig 2. Temporal trend of the number and proportion of DHS surveys with cancer-specific questions (1985–2021).**

most (54 surveys). In this domain, participants were asked in closed-end yes/no questions if they have heard of cervical cancer screening, with specific reference to Pap smear test in most of the such questions. Questions asking if respondents had been screened for cervical cancer in the past 12 months were commonly asked. A minority of surveys included additional questions about the type of cervical screening and the outcome of the screening. Notably, some questionnaires provided information on cancer screening procedures such as Pap smear and visual inspection with acetic acid (VIA), before proceeding to ask respondents if they had been screened for cervical cancer. Questions relating to cervical cancer awareness and knowledge were assessed in 16 surveys. These included questions on whether respondents have ever heard of cervical cancer; their knowledge of the Human papillomavirus (HPV); and signs and symptoms of cervical cancer. In five surveys, respondents were asked if they had cervical cancer as an indicator of prevalence. Only five surveys included questions on breast cancer treatment, by asking participants if they received treatment following a cervical cancer diagnosis.

Questions about barriers to access to cervical cancer services were asked in three surveys, such as those asking why participants did not have a Pap smear; reasons for not receiving screening results where participants reported that they had been screened and reasons for not receiving treatment following a diagnosis of cervical cancer. These were closed-ended questions with response options categorised as personal reasons (lack of time, low awareness of cancer, lack of awareness of cancer service delivery points, health-seeking behaviour, fear and myths), health system-level factors (unavailability of services, distance to a health facility, poor

**Table 1. Cancer-specific questions by country and survey years.**

| Country | Region | Survey year | Surveys with breast cancer questions | Surveys with cervical cancer questions | Surveys with prostate cancer questions | Surveys with questions on other cancers |
|---|---|---|---|---|---|---|
| Angola | Africa | 2016 | None | None | None | None |
| Benin | Africa | 1996, 2001, 2006, 2012, 2018 | None | 2018 | None | 2018 |
| Botswana | Africa | 1988 | None | None | None | None |
| Burkina Faso | Africa | 1993, 1999, 2003, 2010 | 2010 | 2010 | None | None |
| Burundi | Africa | 1987, 2010, 2017 | None | None | None | 2017 |
| Cameroon | Africa | 1991, 1998, 2004, 2011, 2018 | None | 2018 | None | 2018 |
| Cape Verde | Africa | 2005 | None | None | None | None |
| Central African Rep | Africa | 1995 | None | None | None | None |
| Chad | Africa | 1997, 2004, 2015 | None | None | None | None |
| Comoros | Africa | 1996, 2012 | None | None | None | None |
| Congo | Africa | 2005, 2012 | None | None | None | None |
| Congo (Dem Rep) | Africa | 2007, 2014 | None | None | None | None |
| Cote d'Ivoire | Africa | 1994, 1999, 2012 | 2012 | 2012 | None | None |
| Egypt | Africa | 1988, 1992, 1995, 1997, 1998, 2000, 2003, 2005, 2008, 2014 | None | None | None | None |
| Equatorial Guinea | Africa | 2011 | 2011 | 2011 | None | None |
| Eritrea | Africa | 1995, 2002 | None | None | None | None |
| Eswatini | Africa | 2007 | None | None | None | None |
| Ethiopia | Africa | 2000, 2005, 2011, 2016, 2019 | None | None | None | None |
| Gabon | Africa | 2000, 2012 | None | None | None | None |
| Gambia | Africa | 2013, 2020 | None | None | None | None |
| Ghana | Africa | 1988, 1993, 1998, 2003, 2008, 2014 | None | None | None | 2008 |
| Guinea | Africa | 1992, 1999, 2005, 2012, 2018 | None | None | None | None |
| Kenya | Africa | 1989, 1993, 1998, 2003, 2009, 2014 | 2014 | 2014 | 2014 | None |
| Lesotho | Africa | 2004, 2009, 2014 | 2009, 2014 | 2009, 2014 | None | None |
| Liberia | Africa | 1986. 2007, 2013, 2020 | None | None | None | None |
| Madagascar | Africa | 1992, 1997, 2004, 2009, 2021 | None | 2021 | None | None |
| Malawi | Africa | 1992, 2000, 2004, 2010, 2016 | None | None | None | None |
| Mali | Africa | 1987, 1996, 2001, 2006, 2013, 2018 | None | None | None | None |
| Mauritania | Africa | 2001 | None | None | None | None |
| Morocco | Africa | 1987, 1992, 1995, 2004 | None | None | None | 2004 |
| Mozambique | Africa | 1997, 2003, 2011, 2021 | None | None | None | 2021 |
| Namibia | Africa | 1992, 2000, 2007, 2013 | 2013, 2000 | 2013 | 2013 | None |
| Niger | Africa | 1992, 1998, 2006, 2012 | None | None | None | 2012 |
| Nigeria | Africa | 1990, 1999, 2003, 2008, 2013, 2018 | None | None | None | None |
| Rwanda | Africa | 1992, 2000, 2005, 2008, 2010, 2015, 2020 | None | 2020 | None | None |
| Sao Tome and Principe | Africa | 2009 | None | None | None | None |
| Senegal | Africa | 1986, 1993, 1997, 1999, 2005, 2011, 2013, 2014, 2015, 2016, 2017, 2018, 2019 | None | None | None | 2014 |
| Sierra Leone | Africa | 2008, 2013, 2019 | None | None | None | None |
| South Africa | Africa | 1998, 2003, 2016 | None | 2016 | None | 1998, 2003, 2016 |
| Sudan | Africa | 1990 | None | None | None | None |

*(Continued)*

**Table 1.** (Continued)

| Country | Region | Survey year | Surveys with breast cancer questions | Surveys with cervical cancer questions | Surveys with prostate cancer questions | Surveys with questions on other cancers |
|---|---|---|---|---|---|---|
| Tanzania | Africa | 1992, 1996, 1999, 2005, 2010, 2016 | None | None | None | None |
| Togo | Africa | 1988, 1998, 2014 | None | None | None | None |
| Tunisia | Africa | 1988 | None | None | None | None |
| Uganda | Africa | 1989, 1995, 2001, 2006, 2011, 2016 | None | None | None | None |
| Zambia | Africa | 1992, 1996, 2002, 2007, 2014, 2018 | None | None | None | None |
| Zimbabwe | Africa | 1988, 1994, 1999, 2006, 2011, 2015 | None | 2015 | None | None |
| Afghanistan | Asia | 2015 | None | None | None | None |
| Bangladesh | Asia | 1994, 1997, 2000, 2004, 2007, 2011, 2014, 2018 | 2014 | None | None | None |
| Cambodia | Asia | 2000, 2005, 2010, 2014 | None | None | 2015 | None |
| India | Asia | 1993, 1999, 2006, 2016, 2021 | 2016, 2021 | 2016, 2021 | None | 2016, 2021 |
| Indonesia | Asia | 1987, 1991, 1994, 1997, 2003, 2007, 2012, 2017 | None | None | None | None |
| Jordan | Asia | 1990, 1997, 2002, 2007, 2009, 2012, 2018 | 2002, 2007, 2012, 2018 | 2007, 2012, 2018 | None | 2018 |
| Kazakhstan | Asia | 1995, 1999 | None | None | None | None |
| Kyrgyz Republic | Asia | 1997, 2012 | None | None | None | None |
| Maldives | Asia | 2009, 2017 | None | None | None | 2017 |
| Myanmar | Asia | 2016 | None | None | None | None |
| Nepal | Asia | 1996, 2001, 2006, 2011, 2016 | None | None | None | 2006 |
| Pakistan | Asia | 1991, 2007, 2013, 2018 | None | None | None | 2007 |
| Philippines | Asia | 1993, 1998, 2003, 2008, 2013, 2017 | 2017, 2013, 2008, 2003, 1998 | 2017, 2013, 2003, 1998 | 2013, 2003 | 2017, 2013, 2008, 2003, 1998 |
| Sri Lanka | Asia | 1987, 2007, 2016 | None | None | None | None |
| Tajikistan | Asia | 2012, 2017 | 2012 | 2012 | None | None |
| Thailand | Asia | 1987 | None | None | None | None |
| Timor-Leste | Asia | 2010, 2016 | None | 2016 | None | 2016 |
| Turkmenistan | Asia | 2000 | None | None | None | None |
| Uzbekistan | Asia | 1996 | None | None | None | None |
| Vietnam | Asia | 1997, 2002 | None | None | None | None |
| Yemen | Asia | 1992, 1997, 2013 | None | None | None | 2013 |
| Albania | Europe | 2009, 2018 | 2009, 2018 | 2009, 2018 | None | 2009, 2018 |
| Armenia | Europe | 2000, 2005, 2010, 2016 | 2000, 2005, 2010 | 2010, 2005, 2000 | 2010 | 2010 |
| Azerbaijan | Europe | 2006 | None | None | None | None |
| Moldova | Europe | 2005 | None | None | None | None |
| Turkey | Europe | 1993, 1998, 2003, 2008, 2013, 2018 | None | None | None | None |
| Ukraine | Europe | 2007 | None | None | None | None |
| Bolivia | L/America | 1989, 1994, 1998, 2003, 2008 | None | 2008, 2003 | None | None |
| Brazil | L/America | 1986, 1991, 1996 | None | 1986, 1996 | None | 1991 |
| Colombia | L/America | 1996, 1990, 1995, 2000, 2005, 2010, 2015 | 2015, 2010, 2005 | 2015, 2010, 2005, 1990 | 2015 | 2015, 2010 |
| Dominican Republic | L/America | 1986, 1991, 1996, 1999, 2002, 2007a, 2007b, 2013a, 2013b | 2013a, 2013b, 2007, 2002, 1996 | 2013a, 2013b, 2007, 2002, 1996 | None | 2013, 2007, 2002, 1996, 1986 |
| Ecuador | L/America | 1987 | None | None | None | None |
| El Salvador | L/America | 1985 | None | None | None | None |

*(Continued)*

**Table 1.** (Continued)

| Country | Region | Survey year | Surveys with breast cancer questions | Surveys with cervical cancer questions | Surveys with prostate cancer questions | Surveys with questions on other cancers |
|---|---|---|---|---|---|---|
| Guatemala | L/America | 1987, 1995, 1999, 2015 | None | 2015, 1999, 1987 | None | None |
| Guyana | L/America | 2009 | None | None | None | None |
| Haiti | L/America | 1995, 2000, 2006, 2012, 2017 | None | 2017 | None | 2017 |
| Honduras | L/America | 2006, 2012 | 2006, 2012 | 2012, 2006 | 2012 | None |
| Mexico | L/America | 1987 | None | None | None | None |
| Nicaragua | L/America | 1998, 2001 | None | None | None | None |
| Paraguay | L/America | 1990 | None | None | None | None |
| Peru | L/America | 1986, 1992, 1996, 2000, 2006, 2008, 2009, 2010, 2011, 2012, 2013, 2014 | 2009, 2010, 2011, 2013, 2014 | 2009, 2010, 2011, 2013, 2014 | None | 2014, 2013, 2011, 2010 |
| Trinidad and Tobago | L/America | 1987 | None | 1987 | None | None |
| Papua New Guinea | Oceania | 2018 | None | None | None | None |
| Samoa | Oceania | 2009 | None | None | None | None |

quality of services, long waiting times, delay in scheduling appointments) and economic factors (cost of transportation and out-of-pocket cost of cancer services). Overall, the content, scope and depth of the questions varied considerably across countries and survey years. For instance, questions on cancer screening in older surveys tended to be limited to whether or not respondents had been screened, whereas more recent surveys often included additional questions, such as those about the outcome of the screening, whether respondents with positive screening had a diagnostic follow-up and if they received treatment following a diagnosis. Table 2 presents examples of questions and how they are worded across each cancer domain.

## Breast cancer

Questions on breast cancer were asked in 41 survey across 17 countries. Of these, questions on breast cancer screening were by far the most commonly asked (41/41 surveys). These included questions asking if survey participants have heard of breast cancer screening, with mammography being the most commonly mentioned screening modality. There were also questions on the practice of clinical breast examination and breast self-examination. However, questions on whether or not participants who had a mammogram received the result of the mammogram; and outcome of the mammogram on the screening results were less commonly asked. Questions on self-reported breast cancer prevalence; asking if respondents had breast cancer or have been diagnosed with breast cancer, were asked in 3 surveys. Breast cancer awareness and knowledge were assessed in 10 surveys. In such questions, participants were asked if they have ever heard of breast cancer; who can get breast cancer; and signs or symptoms of breast cancer. Five surveys included questions on breast cancer treatment; often asking respondents with cancer if they have had any treatment for cancer following a diagnosis.

Questions about barriers to access to cancer services were asked in three surveys. These included those asking why participants did not have a breast cancer screening; reasons for not receiving mammography results and reasons for not accessing treatment following a diagnosis.

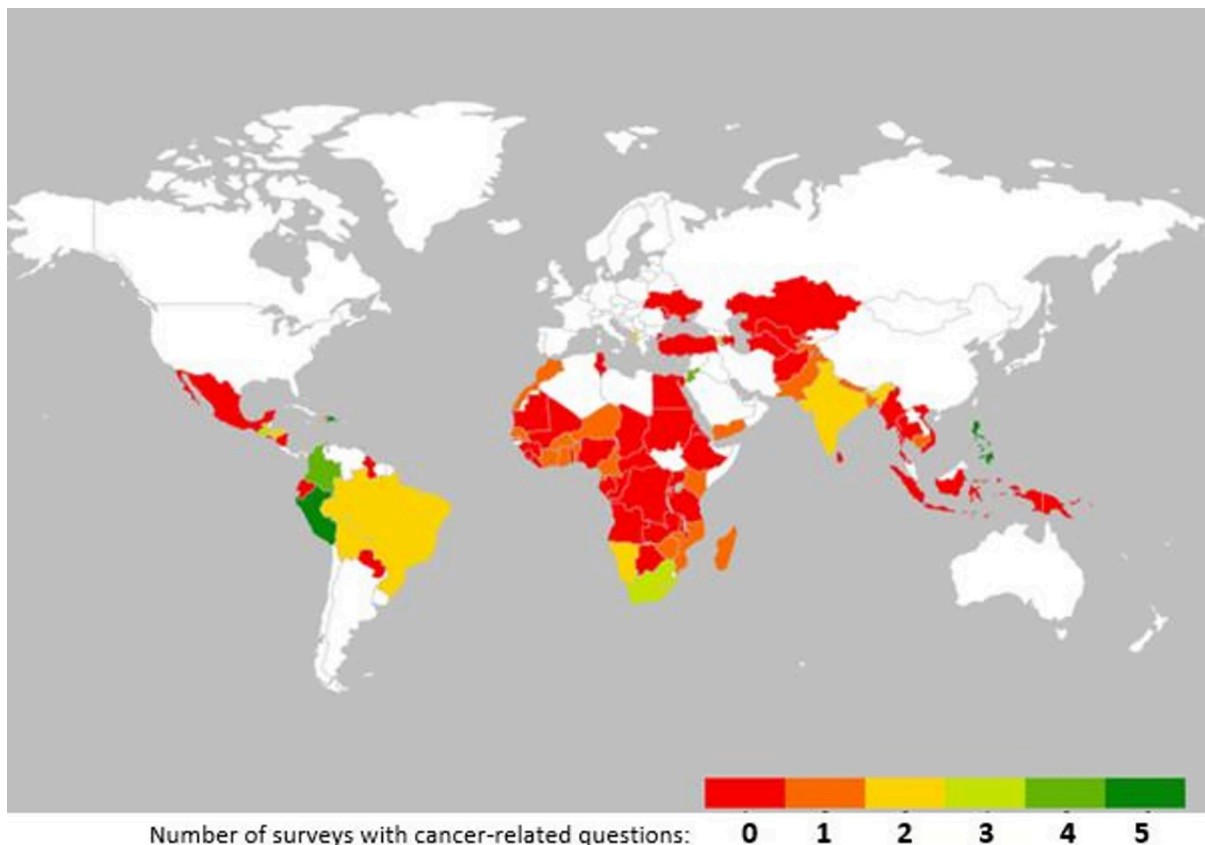

Number of surveys with cancer-related questions:    0   1   2   3   4   5

**Fig 3. Global distribution of DHS surveys with cancer-specific questions by frequency across countries.** Map created in miscrosoft excel using Geographic Heat Map add-in: an open-source tool available via: https://appsource.microsoft.com/en-us/product/office/WA103304320?tab=DetailsAndSupport.

As in the case of cervical cancer, these were closed-ended questions with response options categorised under personal, health system and economic factors. Overall, the content and scope of the cancer questions varies considerably across countries and survey. Table 2 presents examples of questions and how they are worded across each cancer domain.

## Prostate cancer

Prostate cancer questions featured in eight surveys across seven countries. All eight surveys asked respondents if they have had prostate cancer screening. One survey included a question on self-reported prostate cancer prevalence; asking if respondents had prostate cancer or had been diagnosed with prostate cancer. Two surveys included questions on prostate cancer awareness, such as those asking participants if they have ever heard of prostate cancer or know the signs or symptoms of prostate cancer. Another survey included a question on prostate cancer treatment; asking respondents if they have had any treatment for prostate cancer following a diagnosis. One survey asked a question about barriers to access to prostate cancer services. See Table 2 for examples of questions as worded across each cancer domain.

## Other cancers

Questions on other site-specific cancers and non-site specific cancer were included in a total of 40 surveys across 24 countries. These included questions on malignancies such as colorectal,

**Table 2. Cancer-specific questions by county and cancer domains.**

| Cancer type | Cancer domains, number of surveys and example of questions | | | | |
|---|---|---|---|---|---|
| | Cancer prevalence | Awareness & knowledge | Screening and detection | Treatment & follow-up | Barriers to access to cancer services |
| Breast cancer (41 surveys) | 3 surveys<br><br>• Has a doctor or other health professional ever told you that you have breast cancer? | 10 surveys<br><br>• Have you ever heard of breast cancer?<br>• Who can get breast cancer: women only, men only, or both men and women?<br>• What signs or symptoms would lead you to think that a woman has breast cancer? | 41 surveys<br><br>• Have you ever had a breast cancer test?<br>• Have you heard of mammography?<br>• Where did you have your last mammogram?<br>• Has a doctor or other health professional examined your breasts to detect or check for breast cancer?<br>• Have you ever examined your breasts to detect or check for breast cancer?<br>• Have you had a breast cancer clinical exam to detect breast cancer in the last 12 months?<br>• Did you receive the result of the last mammogram?<br>• What was the result of the last mammogram?<br>• Why didn't you claim the result of the last mammogram?<br>• Because of the abnormal mammogram result, did you have to undergo a biopsy? | 3 surveys<br><br>• Did you receive treatment as a result of the mammogram or of the biopsy? | 3 surveys<br><br>• What is the main reason why you have not done mammography?<br>• What was the main reason you did not receive treatment? |
| Cervical cancer (54 surveys) | 5 surveys<br><br>• Has a doctor or other health professional ever told you that you have cancer of the cervix? | 16 surveys<br><br>• Have you heard of cervical cancer?<br>• Have you heard about the Human Papilloma Virus—HPV?<br>• Do you think that the human papillomavirus can cause cervical cancer, also called cervical cancer? | 54 surveys<br><br>• Have you heard of tests for cervical cancer of the uterus?<br>• Have you ever had a cervical cancer test?<br>• In the past 12 months, did you have any cervical cancer screening (Pap smears)?<br>• You received treatment as a result of the Pap smear or colposcopy—biopsy?<br>• What should a woman do when the cytology result is abnormal?<br>• Have you ever had a "Pap" smear to test for cervical cancer?<br>• What type of exam did you have to see if you have cervical cancer?<br>• How long ago was your last test for cervical cancer?<br>• What was the result of your last cervical cancer test?<br>• The last time you had a Pap smear, did you get the result of the test? | 5 surveys<br><br>• Have you had any treatment for the cervix or have you made follow-up visits because of the results of the test? | 3 surveys<br><br>• Why hasn't a pap smear test been taken?<br>• What was the main reason you did not claim the result of the last cytology?<br>• Why have you not sought treatment? |

*(Continued)*

**Table 2.** (Continued)

| Cancer type | Cancer domains, number of surveys and example of questions | | | | |
|---|---|---|---|---|---|
| | Cancer prevalence | Awareness & knowledge | Screening and detection | Treatment & follow-up | Barriers to access to cancer services |
| Prostate cancer (8 surveys) | 1 survey<br><br>• Has a doctor or other health professional ever told you that you have prostate cancer? | 2 surveys<br><br>• Have you ever heard of prostate cancer? | 8 surveys<br><br>• Have you had prostate examination?<br>• Has a doctor or health care professional ever examined you to detect or test for prostate cancer?<br>• Have you ever had a test or exam to see if you have prostate cancer?<br>• Did you have a biopsy or an ultrasound done to determine the type of tumour?<br>• What was the result of the biopsy or the ultrasound? | 1 survey<br><br>• Did you receive medical treatment for the cancer at the time of the diagnosis? | 1 survey<br><br>• Are there any services that you need from a health provider that are not covered by NHIS? (prostate cancer screening listed as an option) |
| Other cancers (40 surveys) | 34 surveys<br><br>• Has a doctor or other health professional ever told you that you have cancer or tumour?<br>• Do you suffer from any of the following diseases? (with options including 'cancer)'.<br>• What type of chronic illness do you have? (with options including 'cancer) | 4 surveys<br><br>• In the last 12 months, have you received educational information on some of the following topics (options included cancer)?<br>• In what ways do you believe smoking can cause health problems? (lung and laryngeal cancer as options)<br>• What signs and symptoms would make you suspect that a person may have cancer? | 8 surveys<br><br>• Has the doctor/other health professional tested you for cancer?<br>• Have you ever been screened for cancer of the oral cavity?<br>• Have you ever been screened for hidden blood in your stool? | 4 surveys<br><br>• Did you receive medical treatment for the cancer at the time of the diagnosis?<br>• Have you sought treatment for this condition?<br>• Are you currently undergoing treatment for cancer? | 1 survey<br><br>• Why have you not sought treatment? |

laryngeal, liver, lung, oral cavity, ovarian and non-site-specific cancers. This aspect likely reflects country-specific cancer burden and priorities other than breast, cervical and prostate cancer; such as questions on oral cavity cancer in India and colorectal cancer in Colombia. Unlike for breast, cervical and prostate cancer for which most questions concerned screening, most of the questions here related to prevalence (34 surveys). These included questions asking respondents if a doctor or other health professional ever told them that they had any cancer or tumour, or if they suffered from any of chronic health conditions (with options including cancer). Questions on cancer screening featured in eight surveys, asking participants in specific age-groups if they have been screened for any cancer (or specific cancers such as cancer of the oral cavity in India, and screening for faecal occult blood for colorectal cancer in Colombia) in the past 12 months.

Questions relating to cancer awareness were included in four surveys. Such questions asked respondents if they received educational information on health topics (options included cancer) and what signs and symptoms would make them suspect that a person may have cancer. Cancer treatment-related questions were asked in four surveys: typically asking respondents if they received treatment for cancer following a cancer diagnosis. Only in one survey were participants asked about barriers to access to cancer services. These included those asking why participants did not have a cancer screening and or receive treatment following a cancer diagnosis (see Table 2).

## Cancer risk factors (alcohol consumption and tobacco use)

Survey questionnaires were also searched for questions relating to known cancer risk factors. The majority of all surveys with available final reports and questionnaires, included questions on alcohol consumption and tobacco use, which are known cancer risk factors. Alcohol-related questions were included in 177 DHS surveys to date. These include questions asking participants if they have ever consumed any alcoholic drink, what type of alcoholic drinks and quantity and frequency of consumption. Questions related to tobacco use were included in 198 surveys. Participants were asked if they currently or ever used tobacco products; what kind of tobacco products were used (including cigarettes, e-cigarettes, smokeless tobacco and chewing tobacco); as well as the quantity and frequency of tobacco use. Notably, these questions were often not asked in the context of cancer, but rather asked in separate sections for substance use. In a few cases, however, participants were asked what ways they believed smoking can cause health problems (with lung and laryngeal cancer as options).

## Discussion

This review examines the cancer-specific data collected as part of the DHS in LMICs from the inception of the surveys to date. Overall, findings demonstrate that, though cancer-specific questions have improved in both frequency and depth over the years, only a minority of surveys have featured these questions to date, while substantial gaps remain in the scope of questions asked. Less than half (43.3%) of the 90 countries have conducted at least one DHS survey with one or more cancer-specific questions, with just a quarter (25.6%) of the surveys including at least one cancer-specific question. The review's findings have implications for the design and implementation of future DHS and similar population-based surveys for collecting useful and context-appropriate data needed to inform cancer control efforts.

In terms of cancer site, cervical cancer questions were the most commonly asked across surveys, followed by those about breast cancer. Questions tended to reflect regional or country-specific cancer burden, such as the preponderance of DHS survey questions on cancer of the cervix in South Africa, colorectal cancer in Colombia and cancer of the oral cavity in India, reflecting the countries' respective incidence, morbidity and mortality burden of these cancers [1]. The relatively higher occurrence of cervical and breast cancer related questions may reflect the growing burden of both cancers in LMICs [15]. Yet, only a minority of DHS surveys have included questions about both cancers. This represents a missed opportunity to utilise large scale population-based surveys like DHS to collect data that can be used to track the burden and trends of these cancers, given that such opportunities are particularly vital to LMICs where cancer registries and health information systems may be weak or lacking [12, 16]. It is thus imperative that DHS surveys are leveraged as opportunities to collect data not only on the burden of cancer, but also on other important aspects of awareness, prevention, screening and early diagnosis and treatment. To achieve this, there is a need for the DHS surveys to develop and standardise a comprehensive set of questions that can be adapted for different contexts. In line with the WHO's mandate on using population-based surveys such as the DHS to drive data-informed decision making for cancer prevention and control, this will help facilitate the collection of data for driving cancer control programme planning and evaluation [17].

In terms of specific cancer care and service delivery domains, questions related to cancer screening were the most commonly featured. The scope and depth of the questions varied widely across countries and survey years. One positive finding was that surveys may provide opportunities for informing and educating people on common cancers, such as by describing cancer signs and symptoms. Though survey participants represent a small proportion of the reference populations, including cancer-specific information and questions in surveys may

contribute to efforts to increase public knowledge and awareness of common cancers. Surveys often included descriptions of screening methods to respondents before proceeding to ask them if they had recently undergone cancer screening. In most cases, however, questions on cancer screening often did not have additional questions to ask which screening methods respondents had or the outcome of the screening. Moreover, survey participants who reported having a recent cancer screening were seldom asked questions about follow-up and treatment. This trend may reflect the current state of cancer screening in LMICs where such services may not be easily accessible or may have weak referral systems to ensure the follow-up and referral of people with positive screening results [15, 18]. As such, current DHS data may only provide information on cancer screening coverage among the target population, but may not offer information needed by countries to evaluate the effectiveness of their cancer screening programmes beyond coverage estimates. To enhance the availability of data for cancer programme planning, monitoring, and evaluation, we recommend that population-based surveys include a core set of questions measuring cancer screening coverage, screening interval, and follow-up and treatment, all of which can help to strengthen cancer surveillance systems.

As countries strengthen and expand access to their national cancer prevention and control programmes, it is important that they pay due attention to the various barriers to access. The promotion of awareness, early detection and access to treatment of cancer is a pillar of any country's comprehensive cancer control strategy [19]. A sound understanding of barriers to cancer services is therefore vital for informing interventions and strategies for addressing those barriers [20]. A few DHS questionnaires have included questions to assess respondents' ability or inability to access cancer services, such as cancer diagnostic and treatment services. DHS surveys thus offer an important tool for assessing barriers to cancer services from nationally-representative samples of participants.

This review found that survey respondents were commonly asked questions on alcohol and tobacco use which are known risk factors of cancer. However, these questions were often not asked in the context of cancer, as they asked in separate sections for substance use. Beyond quantifying the use of these substances, efforts should be made to establish a link between such risk factors and cancer, such as by adding asking additional questions on whether survey participants think the use of these substances is associated with risks of specific cancer types. That linkage can enhance the capacity of cancer control programmes in LMICs to collect and aggregate data on common risk factors and assess their prevalence over time [21]. Such data will be useful for informing and strengthening cancer prevention strategies, while identifying at-risk populations for targeted cancer screening and control initiatives.

While improvements in the frequency and quality of cancer-specific questions in the DHS surveys have many benefits, they have cost and resource implications. Expanding the scope of questions will necessitate the need for additional training of data collectors and will require longer survey time for administer the questionnaires to participants. For this reason and given the resource limitation in most DHS survey contexts, implementers of DHS and similar population-based surveys are likely to consider such resource implications in making decisions on whether to expand the content and scope of survey questionnaires. Nonetheless, it is likely that any cost associated with the expansion of survey questionnaires to accommodate for more useful cancer-specific questions will be offset by the direct and indirect benefits of using the additional data obtained.

Notwithstanding the conceptual and methodological strengths of this review, it has some noteworthy limitations. First, while it used a systematic search strategy and was able to identify all relevant surveys, it is possible that some cancer-specific questions were missed in the search. Secondly, the review focused on the DHS questionnaires and did not assess the broader methodology of the DHS surveys such as by reviewing data collection training manuals and

interview guides for a more nuanced understanding of how questions are asked and phrased in practice. Furthermore, it did not review the conceptualisation of survey questions, nor the use of survey data in practice. Finally, we acknowledge that our review did not comprehensively explore questions on cancer risk factors in the surveys beyond alcohol and tobacco use such as household fuel use, a more comprehensive assessment of which will require a much broader search strategy. These limitations therefore provide opportunities for future reviews on this topic to explore.

## Conclusions

Though cancer-specific questions have been increasingly included in DHS surveys since 1985, only a minority of surveys have featured these questions to date. To aid the collection of more useful population-level data to inform cancer-control priorities and track progress, there is a need to increase the number of surveys asking cancer related questions, while improving the depth and scope of such questions in future DHS surveys.

## Acknowledgments

The authors are grateful to the DHS Programme for making the survey questionnaires and relevant information available for use in this study.

## Author Contributions

**Conceptualization:** Chukwudi A. Nnaji, Jennifer Moodley.

**Data curation:** Chukwudi A. Nnaji, Jennifer Moodley.

**Formal analysis:** Chukwudi A. Nnaji, Jennifer Moodley.

**Methodology:** Chukwudi A. Nnaji, Jennifer Moodley.

**Resources:** Chukwudi A. Nnaji.

**Supervision:** Jennifer Moodley.

**Validation:** Jennifer Moodley.

**Visualization:** Chukwudi A. Nnaji.

**Writing – original draft:** Chukwudi A. Nnaji.

**Writing – review & editing:** Chukwudi A. Nnaji, Jennifer Moodley.

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
