## [Decision Letter · Decision Letter 0]

14 Mar 2023

PGPH-D-22-01991

Collection of cancer-related data in population-based surveys in low- and middle-income countries: a review of the demographic and health surveys

Dear Dr. Nnaji,

Thank you for submitting your manuscript to PLOS Global Public Health. After careful consideration, we feel that it has merit but does not fully meet PLOS Global Public Health’s publication criteria as it currently stands. Therefore, we invite you to submit a revised version of the manuscript that addresses the points raised during the review process.

We look forward to receiving your revised manuscript.

Kind regards,

Samiratou Ouédraogo, DPharm, MPH, Ph.D.

Academic Editor

Journal Requirements:

2. Please send a completed 'Competing Interests' statement, including any COIs declared by your co-authors. If you have no competing interests to declare, please state "The authors have declared that no competing interests exist". Otherwise please declare all competing interests beginning with the statement "I have read the journal's policy and the authors of this manuscript have the following competing interests:"

3. Please provide separate figure files in .tif or .eps format only and remove any figures embedded in your manuscript file. Please also ensure that all files are under our size limit of 10MB.

Additional Editor Comments (if provided):

Reviewers' comments:

Reviewer's Responses to Questions

**Comments to the Author**

1. Does this manuscript meet PLOS Global Public Health’s publication criteria? Is the manuscript technically sound, and do the data support the conclusions? The manuscript must describe methodologically and ethically rigorous research with conclusions that are appropriately drawn based on the data presented.

Reviewer #1: Yes

Reviewer #2: Yes

2. Has the statistical analysis been performed appropriately and rigorously?

Reviewer #1: N/A

Reviewer #2: Yes

3. Have the authors made all data underlying the findings in their manuscript fully available (please refer to the Data Availability Statement at the start of the manuscript PDF file)?

Reviewer #1: No

Reviewer #2: No

4. Is the manuscript presented in an intelligible fashion and written in standard English?

Reviewer #1: Yes

Reviewer #2: Yes

5. Review Comments to the Author

Reviewer #1: The authors have done commendable job in examining the cancer-related questions in DHS surveys conducted so far. The research paper is timely and provides useful policy implications for data collection in LMICs, which lack quality data from cancer registries. I have few comments and suggestions for the authors to improve this manuscript.

1. There are few language issues such as " A filtered was ....." in the Methods section.

2. "Peaking at 100%..." What exactly is peaking, please clear the sentence a little more.

3. Results Section- There are few grammatical errors or typos

i) Cervical Cancer: ..

in the past 12 months where. I think it is in "in the past 12 months were".

"In five surveys, respondents were if...."

ii) Breast Cancer

" were asked if they have you ever....

iii) Prostate Cancer

" if they have you ever heard....

iv) Other Cancers

" .... professions ever told them they.... It should be " ... professionals ever told them that they ...

4. DISCUSSION SECTION

.... or may have week. It should be ... or may have weak

5. Why the authors think few questions would have been missed

6. It is a suggestion to include the data on cancer incidence or mortality rate using GLOBOCAN 2020 to shed light on whether the surveyed countries have low or high incidence or mortality rates of cancers for which questions were asked in DHS.

7. It is a suggestion to include geogrphical heatmap of the countries in which DHS survey data is available showing geographically where there is no DHS survey or no cancer related questions to the countries with maximum surveys or maximum cancer related questions.

Reviewer #2: The Demographic and Health Survey gathers a variety of health-related information. While DHS surveys are not designed for cancer surveillance, they can give valuable information for cancer registries, particularly in LMICs where robust cancer registries are lacking. This review provides a concise summary of how cancer questions have been incorporated into DHS. This review is strengthened by the inclusion of countries from different continents and multiple years. This article is well-written and has excellent word flow.

Minor comments.

1. On Page 7: The sentence starting '"Ultimately, findings......would fit in the discussion section.

2. The method section is long, would benefit from subsection e.g Study design, DHS survey structure, Search Strategy and data collection, Data analysis.

3.Minor grammatical errors to correct on page 16 , were vs where,..In five surveys,respondents were *Asked* .Only five surverys included questions on *cervical* cancer treatment.... page 17...if they have ever heard.....

6. PLOS authors have the option to publish the peer review history of their article (what does this mean?). If published, this will include your full peer review and any attached files.

**Do you want your identity to be public for this peer review?** For information about this choice, including consent withdrawal, please see our Privacy Policy.

Reviewer #1: No

Reviewer #2: No

---

## [Decision Letter · Decision Letter 1]

4 Jul 2023

PGPH-D-22-01991R1

Collection of cancer-related data in population-based surveys in low- and middle-income countries: a review of the demographic and health surveys

Dear Nnaji,

Thank you for submitting your manuscript to PLOS Global Public Health. After careful consideration, we feel that it has merit but does not fully meet PLOS Global Public Health’s publication criteria as it currently stands. Therefore, we invite you to submit a revised version of the manuscript that addresses the points raised during the review process. Kindly make sure to address the major comments about definition of outcomes as raised by the reviewer.

We look forward to receiving your revised manuscript.

Kind regards,

Prabhdeep Kaur, DNB Medicine, MAE (Epidemiology)

Academic Editor

Journal Requirements:

1. Please amend your online Financial Disclosure statement. If you did not receive any funding for this study, please simply state: “The authors received no specific funding for this work.”

Additional Editor Comments (if provided):

Reviewers' comments:

Reviewer's Responses to Questions

**Comments to the Author**

1. If the authors have adequately addressed your comments raised in a previous round of review and you feel that this manuscript is now acceptable for publication, you may indicate that here to bypass the “Comments to the Author” section, enter your conflict of interest statement in the “Confidential to Editor” section, and submit your "Accept" recommendation.

Reviewer #2: All comments have been addressed

Reviewer #3: (No Response)

2. Does this manuscript meet PLOS Global Public Health’s publication criteria? Is the manuscript technically sound, and do the data support the conclusions? The manuscript must describe methodologically and ethically rigorous research with conclusions that are appropriately drawn based on the data presented.

Reviewer #2: Yes

Reviewer #3: Yes

3. Has the statistical analysis been performed appropriately and rigorously?

Reviewer #2: Yes

Reviewer #3: N/A

4. Have the authors made all data underlying the findings in their manuscript fully available (please refer to the Data Availability Statement at the start of the manuscript PDF file)?

Reviewer #2: No

Reviewer #3: Yes

5. Is the manuscript presented in an intelligible fashion and written in standard English?

Reviewer #2: Yes

Reviewer #3: Yes

6. Review Comments to the Author

Reviewer #2: All my comments have been responded to

Reviewer #3: The paper presents an interesting description of cancer related data being collected in DHS globally. DHS do represent a potentially valuable source for NCD surveillance data including on risk factors, screening and treatment. An evaluation is therefore worthwhile.

Major Issues:

My main comment would be related to a clear definition of “cancer -related data”. It has not been clearly defined. This being the main outcome variable, needs to be defined. For example cancer related risk factors are covered in the paper, as a peripheral issue. Number of sexual partners is a known behavioural risk factor for cervical cancer, oral pill use has implications for cancer. Household fuel use also fits the bill as a risk factor for cancer and is usually collected in DHS.

Search strategy does not list them though (line 150) says in the third step it was included? Not sure how anything that is not included in first step can be included in third step.

Either the title and scope of the paper should be changed, or the definition revised. Either way a clear definition is a MUST.

Minor issues:

Table 1 can be revised to leave out the countries that do not have any survey on cancer related data. These countries can be added as a legend (with years of survey in brackets) to make the table more reader friendly. Else it is made as a supplementary table. Fig 4 provides the same information.

Figure 2 & 3 can be combined and rather than individual years can be given in blocks of 3 years. It gives a wrong impression when the authors say that in 2021 100% of the surveys asked cancer questions as the number of surveys is 2 and in the previous two years (2019-20), it was quite low. It can be expressed either as numbers or percentages. The figure was not clear in black and white.

It is not clear how the authors suggestion of linking RF questions with cancer related section (line 362-366). Authors need to explain this more.

336-338. I would say that surveys provide an opportunity to educate people as the population covered is miniscule as compared to the whole population.

7. PLOS authors have the option to publish the peer review history of their article (what does this mean?). If published, this will include your full peer review and any attached files.

**Do you want your identity to be public for this peer review?** For information about this choice, including consent withdrawal, please see our Privacy Policy.

Reviewer #2: No

Reviewer #3: **Yes: **Dr. Anand Krishnan

---

## [Editor Report · Decision Letter 2]

7 Aug 2023

Collection of cancer-specific data in population-based surveys in low- and middle-income countries: a review of the demographic and health surveys

PGPH-D-22-01991R2

Dear Dr Nnaji

We are pleased to inform you that your manuscript 'Collection of cancer-specific data in population-based surveys in low- and middle-income countries: a review of the demographic and health surveys' has been provisionally accepted for publication in PLOS Global Public Health.

Best regards,

Prabhdeep Kaur, DNB Medicine, MAE (Epidemiology)

Academic Editor